# Acetyl-CoA carboxylase 1 and 2 inhibition ameliorates steatosis and hepatic fibrosis in a MC4R knockout murine model of nonalcoholic steatohepatitis

**Mitsuharu Matsumoto[1]\***, **Hiroaki Yashiro[2]**, **Hitomi Ogino[1]**, **Kazunobu Aoyama[3]**, **Tadahiro Nambu[4]**, **Sayuri Nakamura[4]**, **Mayumi Nishida[1]**, **Xiaolun Wang[2]**, **Derek M. Erion[2]**, **Manami Kaneko[1]**

1 Department of Integrated Biology, Axcelead Drug Discovery Partners, Inc., Fujisawa, Kanagawa, Japan, 2 Gastroenterology Drug Discovery Unit, Takeda Pharmaceutical Company Limited, Cambridge, Massachusetts, United States of America, 3 Department of Drug Disposition & Analysis, Axcelead Drug Discovery Partners, Inc., Fujisawa, Kanagawa, Japan, 4 Department of Nonclinical Safety Research, Axcelead Drug Discovery Partners, Inc., Fujisawa, Kanagawa, Japan

\* mitsuharu.matsumoto@axcelead.com

**Data Availability Statement:** All relevant data are within the manuscript and supplemental figure.

## Abstract

Acetyl-CoA carboxylase (ACC) catalyzes the rate-limiting step in *de novo* lipogenesis, which is increased in the livers of patients with nonalcoholic steatohepatitis. GS-0976 (firsocostat), an inhibitor of isoforms ACC1 and ACC2, reduced hepatic steatosis and serum fibrosis biomarkers such as tissue inhibitor of metalloproteinase 1 in patients with nonalcoholic steatohepatitis in a randomized controlled trial, although the impact of this improvement on fibrosis has not fully been evaluated in preclinical models. Here, we used Western diet-fed melanocortin 4 receptor-deficient mice that have similar phenotypes to nonalcoholic steatohepatitis patients including progressively developed hepatic steatosis as well as fibrosis. We evaluated the effects of ACC1/2 inhibition on hepatic fibrosis. After the confirmation of significant hepatic fibrosis with a 13-week pre-feeding, GS-0976 (4 and 16 mg/kg/day) treatment for 9 weeks lowered malonyl-CoA and triglyceride content in the liver and improved steatosis, histologically. Furthermore, GS-0976 reduced the histological area of hepatic fibrosis, hydroxyproline content, mRNA expression level of type I collagen in the liver, and plasma tissue metalloproteinase inhibitor 1, suggesting an improvement of hepatic fibrosis. The treatment with GS-0976 was also accompanied by reductions of plasma ALT and AST levels. These data demonstrate that improvement of hepatic lipid metabolism by ACC1/2 inhibition could be a new option to suppress fibrosis progression as well as to improve hepatic steatosis in nonalcoholic steatohepatitis.

## Introduction

The prevalence of nonalcoholic fatty liver disease (NAFLD) is rapidly increasing worldwide and now the most common liver disorder in the western world [1]. NAFLD is strongly

**Funding:** This study was funded by Takeda Pharmaceutical Company limited. During the time this study was conducted, Takeda provided support in the form of salaries for authors [HY, XW, DE]. Axcelead Drug Discovery Partners, Inc., also provided salaries for authors [MM, HO, KA, TN, SN, MN, MK], but did not fund this study. These companies did not have any additional role in the study design, data collection and analysis, decision to publish or preparation of the manuscript. The specific roles of these authors are articulated in the 'author contributions' section.

**Competing interests:** There is no competing interest exists. The commercial affiliation between Takeda and Axcelead dose not alter our adherence to PLOS ONE policies on sharing data and materials.

associated with metabolic abnormalities such as obesity, insulin resistance, and type 2 diabetes mellitus, and it encompasses complicated and extensive liver diseases including asymptomatic steatosis and more aggressive nonalcoholic steatohepatitis (NASH) [2, 3]. NASH is characterized by steatosis, cytoskeletal damage (hepatocellular ballooning), lobular inflammation and fibrosis [4]. Because the progression of fibrosis in NASH leads to liver cirrhosis, which results in liver failure, portal hypertension, and hepatocellular carcinoma, suppressing the progression of fibrosis is critical in order to improve mortality rates [5, 6]. Despite the significant efforts in clinical development for drug to treat fibrosis and NASH, there is currently no approved drugs to treat these conditions [7, 8]. Therefore, available therapy for NASH is limited mostly to lifestyle interventions such as weight loss and vitamin E supplementation [9].

The pathogenesis of NASH has been an area of intense interest in recent research and development. The multiple parallel hits theory comprises a wide spectrum of potential risk factors such as insulin resistance, oxidative stress, proinflammatory cytokines and microbiota changes [10]. Among these hypotheses, it has been consistently demonstrated that insulin resistance plays an important role in the progression of NASH, following numerous studies in animal models and patients with NAFLD [11]. In insulin-resistant states, hyperinsulinemia induces an elevation of sterol regulatory element binding protein 1 (SREBP-1) expression in hepatocytes, resulting in the transcriptional activation of all lipogenic genes including ACC which promotes *de novo* lipogenesis (DNL). ACC exists as two isozymes that are encoded by separate genes and display distinct cellular distributions [12–17]. Although both isoforms are expressed in various tissues, ACC1 is predominantly expressed in lipogenic tissues such as liver and adipose tissue, while ACC2 is predominantly expressed in the heart, skeletal muscle and liver. ACC1 is cytosolic while ACC2 is associated with mitochondria. In the liver, malonyl-CoA formed by ACC1 in the cytoplasm is primarily used for DNL, whereas malonyl-CoA formed by ACC2 at the mitochondrial surface allosterically suppresses carnitine palmitoyltransferase I (CPT1) and mitochondrial fatty acid oxidation [12, 15–17]. About 25% of hepatic triglyceride accumulated in patients with NAFLD is derived from hepatic DNL and patients with NAFLD have significantly higher rates of hepatic DNL compared with lean individuals [18,19]. Based on these insights, it appears that triglyceride derived from DNL makes a major contribution to hepatic steatosis; therefore, the inhibition of DNL might be the best approach to suppress or prevent further deterioration of hepatic steatosis [19, 20].

Liver-specific ACC1 knockout (KO) mice, which were independently produced for two different studies, showed a reduction of hepatic DNL compared to that of control mice only when they were fed a high-sucrose diet [21, 22]. Hepatocytes of liver specific ACC2 KO mice displayed higher fatty acid oxidation accompanied with higher CPT1 activity [23]. Furthermore, antisense oligonucleotide inhibitors of ACC2 enhanced fatty acid oxidation in primary rat hepatocytes [24]. These results suggest that dual inhibition of ACC1/2 in hepatocytes has more potential for improvement of steatosis than inhibition of either gene individually, due to both inhibition of DNL and stimulation of fatty acid oxidation [12]. Recently, GS-0976 (firsocostat), a liver-targeted potent inhibitor of both ACC1/2, has been reported to inhibit human ACC1 and ACC2 with $IC_{50}$ values of 2.1 and 6.1 nM, respectively [25]. This compound has been shown to reduce DNL in cultured HepG2 cells, stimulate fatty acid oxidation in cultured C2C12 cells, and reduce hepatic DNL in normal rats [25, 26]. Long-term treatment with GS-0976 reduced hepatic steatosis and improved dyslipidemia in rats with diet-induced obesity. Furthermore, this drug improved hepatic steatosis and lowered hemoglobin A1c in Zucker diabetic rats. GS-0976 also reduced levels of a hepatic steatosis and serum fibrosis marker, tissue metalloproteinase inhibitor 1 (TIMP-1), in patients with NASH [27]. However, in the same study, GS-0976 did not significantly decrease levels of two other serum fibrosis markers,

procollagen III N-terminal peptide and hyaluronic acid. It has not been verified whether inhibition of ACC1/2 would improve hepatic fibrosis in NASH.

In this study, we examined effects of ACC1/2 dual inhibition by GS-0976 on fibrosis using melanocortin 4 receptor (MC4R) KO mice fed a Western diet (WD). MC4R is a seven-transmembrane G-protein-coupled receptor expressed in hypothalamic nuclei and is implicated in the regulation of appetite and body weight [28]. WD-fed MC4R KO mice are known to exhibit pathophysiological changes of NASH including hepatic steatosis, liver fibrosis and hepatocellular carcinoma following the obesity-related phenotype [29, 30]. We generated MC4R KO mice and evaluated the effects of GS-0976 on hepatic steatosis and fibrosis in WD-fed MC4R KO mice. Furthermore, because ACC inhibition has been reported to reduce hepatic steatosis but elevate plasma triglyceride concentrations in mice, rats and patients with NASH [20, 31], we also monitored its influence on plasma parameters to investigate the usefulness of ACC dual inhibition.

## Materials and methods

### Compounds

1,4-dihydro-1-[(2R)-2-(2-methoxyphenyl)-2-[(tetrahydro-2H-pyran-4-yl)oxy]ethyl]-α,α,5-trimethyl-6-(2-oxazolyl)-2,4-dioxo-thieno[2,3-d]pyrimidine-3(2H)-acetic acid, GS-0976, was synthesized as reported previously [25].

### Generation of MC4R knockout mice

A targeting vector for homologous recombination was constructed by insertion of an mCherry unit and a neomycin resistant unit between the transcription start site and the initiation codon of the *MC4R* gene with BAC clone RP23-112M22 using the Red/ET recombination kit (Gene Bridges GmbH, Land Baden-Württemberg, Germany). The resulting vector was electroporated into C57BL/6J mouse ES cells [32] and recombinant cells were selected using G418. The ES cells showing correct homologous recombination were screened by real-time PCR genotyping. The resulting ES cells were injected into ICR mouse tetraploid blastocysts. Chimeric offspring were identified by coat color. Chimeric male mice with high ES cell contribution were crossed with C57BL/6J females and germ line transmission was predicted by coat color and confirmed by PCR genotyping (Fig 1). PCR primer sets for wild type (WT) (P1, 5´-GCAG TACAGCGAGTCTCAGG-3´ and P2, 5´-CTCATAGCATCCTCCGTCCG -3´; 474

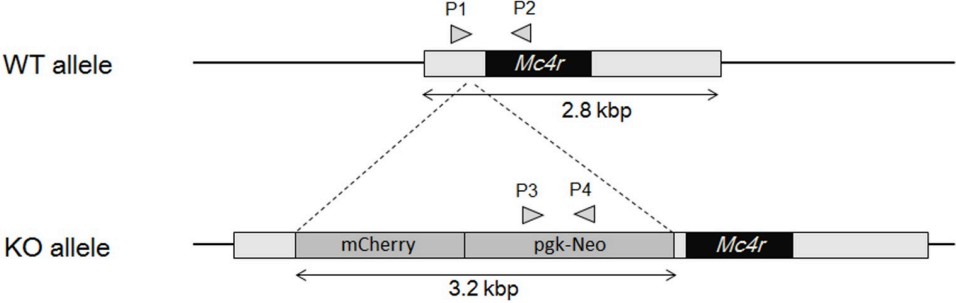

**Fig 1. Schematic diagrams of wild-type *Mc4r* allele and *Mc4r* KO allele.** mCherry unit (mCherry and polyoma polyA signal) and neomycin resistant unit (pgk promoter, neomycin resistant gene and bgh polyA signal) were inserted by homologous recombination between the transcription start site and the initiation codon of the *Mc4r* gene to disrupt *Mc4r* transcription. WT: wild type, KO: knockout, Neo: neomycin resistant unit, pgk: mouse phosphoglycerate kinase, bgh: bovine growth hormone. The primers for PCR genotyping are shown by arrowheads (P1, P2, P3 and P4).

bp) and KO (P3, 5´-GGATCTCCTGTCATCTCACCTTGC-3´ and P4, 5´-TAGCC
AACGCTATGTCCTGATAGC-3´; 374 bp) were used for genotyping.

## Repeated dosing study in WD-fed MC4R knockout mice

Male MC4R KO mice were fed with WD (D12079B; Research Diets, New Brunswick, Canada) for 13 weeks starting at 11 weeks of age. Normal chow-fed (CE-2; CLEA Japan, Tokyo, Japan) wild type littermates of the same age were used as lean controls. Both types of mice were allowed *ad libitum* access to food and water. The mice were individually housed under controlled temperature, humidity and a 12-hour light-dark cycle (lights on 7:00–19:00). Blood was collected from the tail vein. Plasma alanine transaminase (ALT), aspartate aminotransferase (AST), triglyceride, total cholesterol, and glucose were measured enzymatically using a Clinical Analyzer 7180 (Hitachi High-Technologies, Tokyo, Japan). Plasma insulin concentrations were measured using an ultrasensitive mouse insulin enzyme-linked immunosorbent assay kit (Morinaga Institute of Biological Science, Kanagawa, Japan). Plasma TIMP-1 concentrations were measured using Mouse TIMP-1 Quantikine ELISA Kit (R&D Systems, Minnesota, US). Mice were randomly divided into groups based on plasma parameters, the AST/ALT ratio, food intake and body weight. Six MC4R KO mice, whose plasma parameters and body weights were almost identical to the initial values of the individuals selected for repeated dosing study, were euthanized in order to evaluate hepatic triglyceride and hydroxyproline content without drug treatment. GS-0976 (4 and 16 mg/kg/day) was orally administered twice a day for 9 weeks. Body weight and food intake were monitored for 8 weeks from the beginning of treatment with GS-0976. After 8 weeks of treatment, plasma parameters were measured again. After 9 weeks of treatment, all mice were anesthetized with isoflurane (3–5%) 1 hour after the last drug administration and then the liver was harvested for histopathological and gene expression analysis in the same manner. Hepatic triglyceride, hydroxyproline and malonyl-CoA levels were also measured. All animal experiments were approved by the Institutional Animal Care and Use Committee of Shonan Research Center, Takeda Pharmaceutical Company Limited (AU-00011733, AU-00020014, AU-00010116).

## Measurement of tissue malonyl-CoA content

Seven-week-old male C57BL/6J mice fed with normal chow were divided into 7 groups based on their body weights. GS-0976 was orally administered once, then liver and muscle were harvested and frozen 1 hour later. The liver was also harvested 1 hour after the last drug administration in the repeated dosing study using WD-fed MC4R KO mice. These samples were homogenized in 6% perchloric acid containing malonyl-CoA $^{13}C_3$ (Sigma-Aldrich, Missouri, US) as an internal standard. After centrifugation, the supernatant was subjected to solid phase extraction using the Oasis HLB Extraction Cartridge (Waters, Massachusetts, US). The analyte was eluted with acetonitrile supplemented with dibutylammonium acetate (Tokyo Chemical Industries, Tokyo, Japan), followed by rinsing of the column with ultrapurified water. The eluate was dried down under a stream of nitrogen and the residue was reconstituted in 100 μL of ultrapure water. An aliquot of 10 μL was injected into an LC-MS/MS system. The LC-MS/MS setup consisted of a Shimadzu LC-20AD HPLC system (Shimadzu, Kyoto, Japan) and an API5000 mass spectrometer (AB Sciex, California, US). The analytical column was a CAP-CELL CORE C18 (2.7 μm, 2.1 x 50 mm, Shiseido, Kanagawa, Japan) used at 40 ˚C. The mobile phases were composed of (A) 50 mmol/L ammonium carbonate/ammonium hydroxide (pH 9) supplemented with dibutylammonium acetate and (B) acetonitrile. The stepwise gradient program used was as follows: 0–2.5 min, B 1–30%; 2.5–3 min, B 30–95%; 3–4 min, B 95%;

4–4.01 min, B 95–1%; and 4.01–6 min, B 1%. The flow rate of the mobile phase was set at 0.5 mL/min.

## Measurement of hepatic triglyceride and hydroxyproline content

For the measurement of hepatic triglyceride, samples of liver were homogenized at a concentration of 100 mg of tissue per 1 mL of saline, and then the homogenate was mixed thoroughly with a combination of hexane and 2-propanol (3:2). After centrifugation, the upper organic layer containing lipids was collected. Hexane and 2-propanol solution were added, and the upper layer was collected again. The collected upper layers were dried, and the residue was dissolved in 2-propanol. Triglyceride concentration was measured using the Triglyceride-E test (Fujifilm Wako Pure Chemical Industries, Osaka, Japan). Hepatic hydroxyproline content was measured by a commercially available total collagen kit (QuickZyme Biosciences, Zuid-Holland, Netherlands) according to the manufacturer's instructions.

## Histological analysis

Excised liver sections were fixed with 10% neutralized formalin and embedded in paraffin. Three-micrometer paraffin sections were stained with Hematoxylin and Eosin and evaluated NAFLD Activity Score by the pathologists [33]. The pathologists were blinded to the animal information including strain, compound and dose. To evaluate fibrosis, 3-millimeter paraffin sections were stained with 0.1% Sirius Red / 0.1% Fast Green FCF solution. Whole slide digital images were acquired with a Scanscope XT (Leica Microsystems, Tokyo, Japan). The regions of interest were manually drawn passing over the connective tissues around the large blood vessels in a blinded fashion and the percentage of Sirius Red-positive area in the total region of interest was evaluated using ImageScope (v12.3.2.8013, Leica Microsystems).

## mRNA expression analysis by real-time PCR

Total RNA was isolated from 50–100 mg of liver tissue using the RNeasy Lipid Tissue Mini kit (Qiagen, Tokyo, Japan) followed by reverse transcription using the High Capacity RNA-to-cDNA kit (Thermo Fisher Scientific, Tokyo, Japan) according to the manufacturer's instructions. cDNA was amplified by TaqMan Universal Master Mix II (Thermo Fisher Scientific, Tokyo, Japan) using an ABI7900 (Thermo Fisher Scientific, Tokyo, Japan) according to the manufacturer's instructions. Commercially available primer-probe sets were used (Thermo Fisher Scientific, Tokyo, Japan). The sets were as follows: mouse collagen type1 alpha1 (*Col1a1*) (Mm00801666), mouse collagen type1 alpha2 (*Col1a2*) (Mm00483888), *F4/80* (Mm00802529), transforming growth factor-β1 (*TGF-β1*) (Mm01178820). *36B4* (Mm00725448) was used as an endogenous control gene. Relative mRNA expression was calculated by the ΔΔCt method.

## Statistical analysis

All data in the graph are represented as the mean + SD. For evaluation of the effects of GS-0976, statistical differences between vehicle and GS-0976 treatment in MC4R KO mice were analyzed by a one-tailed Williams' test or Shirley-Williams test. The p-values $< 0.025$ were considered statistically significant. To confirm the establishment of the disease state, statistical differences between lean control mice and vehicle-treated WD-fed MC4R KO mice were analyzed by Student's *t*-test or Aspin-Welch test. The p-values $< 0.05$ were considered statistically significant.

## Results

### Tissue malonyl-CoA reduced by a single administration of GS-0976 in normal mice

In order to confirm target engagement of GS-0976 *in vivo*, tissue content of malonyl-CoA, which is a product of ACC1/2, was measured in normal C57BL/6 mice. A single oral administration of GS-0976 (0.3–100 mg/kg) significantly decreased hepatic malonyl-CoA content in a dose-dependent manner (Fig 2). In contrast, reductions of malonyl-CoA by GS-0976 were weaker in skeletal muscle. Although GS-0976 at 30 and 100 mg/kg significantly decreased malonyl-CoA in the skeletal muscle by 88 and 99%, respectively, lower dosing of GS-0976 did not decrease it, suggesting that GS-0976 is liver-specific at single doses of 10 mg/kg or less. Therefore, doses of 2 and 8 mg/kg, b.i.d. (4 and 16 mg/kg/day) were selected for the repeated dosing study to examine the anti-NASH effect of GS-0976 in MC4R KO mice.

### Chronic treatment in WD-fed MC4R KO mice

The experimental protocol for the multiple dosing study in the MC4R KO mice is shown in Fig 3. The body weight of WD-fed MC4R KO mice was 1.8 times greater than that of lean control mice before treatment with GS-0976 (Fig 4A). Body weight in WD-fed MC4R KO mice treated with vehicle was further increased by 8.9% from the initial values (Fig 4C). Cumulative caloric intake of vehicle-treated MC4R KO mice was 1.5 times higher compared to control mice (Fig 4B), indicating that WD-fed MC4R KO mice had an obesity phenotype due in part to overeating as previously reported [33]. Treatment with GS-0976 (4 and 16 mg/kg/day, b.i.d.) for 9 weeks was well tolerated. At 16 mg/kg/day, weight gain was significantly suppressed at 4.5% compared with the vehicle-treated group, without an effect on food intake. Significant effects on weight gain and calorie intake were not observed with GS-0976 at 4 mg/kg/day.

### Blood biochemistry

At the start of drug treatment, WD-fed MC4R KO mice displayed higher plasma ALT and AST, which are released from injured hepatocytes into circulation (Fig 5). Additionally, although the mice maintained normoglycemia, they developed hyperinsulinemia (Table 1). Plasma triglyceride concentrations in MC4R KO mice were similar with those of lean control mice, meanwhile total cholesterol concentrations were 4 times higher compared with those of control mice. These data suggest that WD-fed MC4R KO mice exhibited hepatocellular injury due to metabolic dysfunction from insulin resistance and disruption of lipid metabolism. Treatment with GS-0976 at 4 and 16 mg/kg/day lowered plasma ALT levels by 76 and 82% and AST levels by 70 and 80% compared with vehicle treatment, respectively (Fig 5). GS-0976 at 4 and 16 mg/kg/day also significantly lowered plasma total cholesterol concentrations by 32 and 36% compared with vehicle treatment, respectively. At the same time, GS-0976 at 4 and 16 mg/kg/day significantly increased plasma insulin concentrations 6- and 8-fold, and slightly but significantly increased plasma glucose concentrations 1.1- and 1.2-fold compared with the vehicle-treated group, respectively. Furthermore, both doses significantly increased plasma triglyceride concentrations compared with vehicle treatment. These data suggest that GS-0976 improved markers of hepatic injury and cholesterol metabolism, whereas this compound deteriorated glucose and triglyceride metabolism and accelerated hyperinsulinemia.

### Liver weights and hepatic triglyceride content

Liver weights and hepatic triglyceride content in vehicle-treated MC4R KO mice after 9-week treatment were 4.3- and 12-fold higher than those in lean control mice (Fig 6). The liver

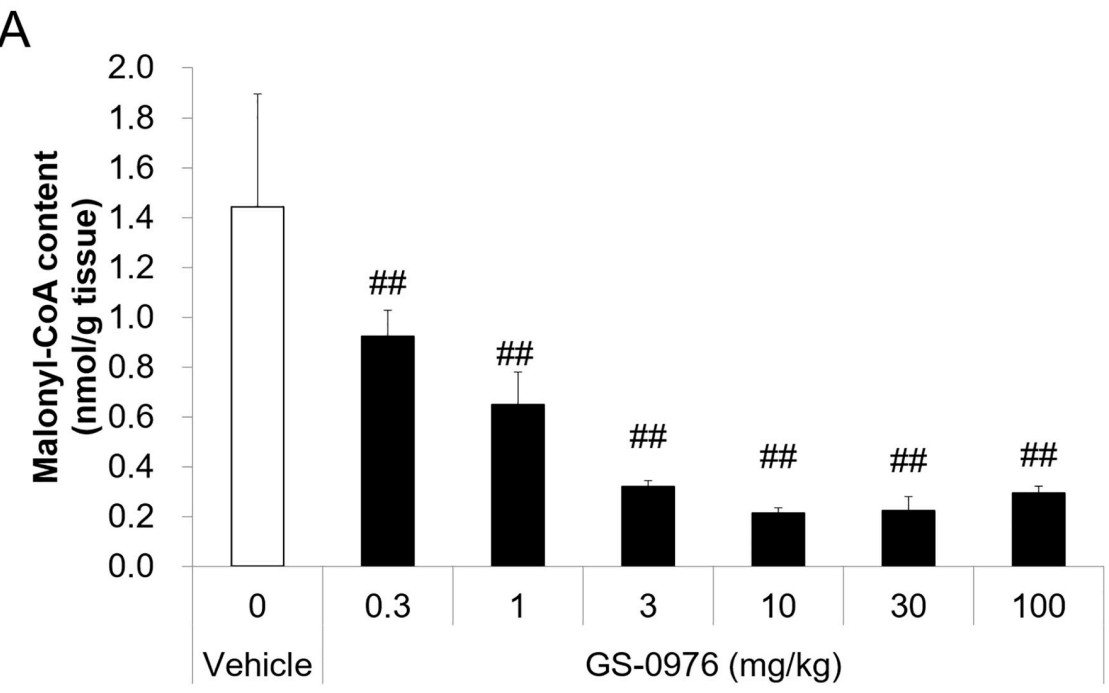

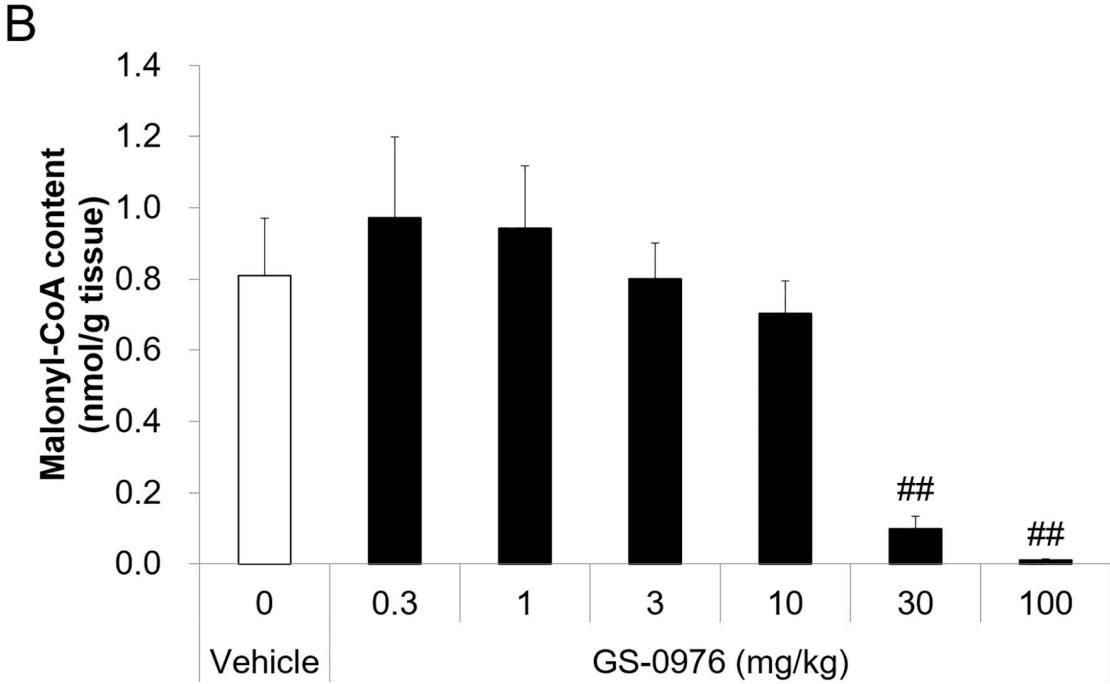

**Fig 2. Effects of a single oral administration of GS-0976 on liver and skeletal muscle malonyl-CoA content in C57BL/6 mice.**
(A) Liver malonyl-CoA content. (B) Gastrocnemius skeletal muscle malonyl-CoA content. The liver and skeletal muscle were harvested 1 hour after a single oral administration of vehicle (n = 10) or GS-0976 (0.3–100 mg/kg, n = 5) from C57BL/6J mice fed with normal chow. Data are represented as the mean + SD, ##$p < 0.005$ vs. vehicle by one-tailed Shirley-Williams test.

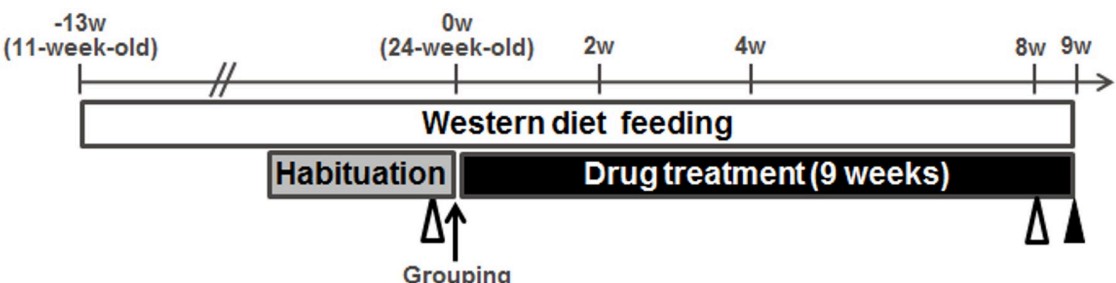

**Fig 3. Experimental protocol for repeated dosing study using MC4R KO mice.** Eleven-week-old male MC4R KO mice were fed with WD for 22 weeks. Drug treatments were started 13 weeks after pre-feeding. GS-0976 (4 and 16 mg/kg/day) was orally administered twice a day for 9 weeks starting at 24 weeks of age. The white triangle marks the collection of blood before and after the 8-week treatment. The black triangle marks harvesting of the liver after 9 weeks of treatment.

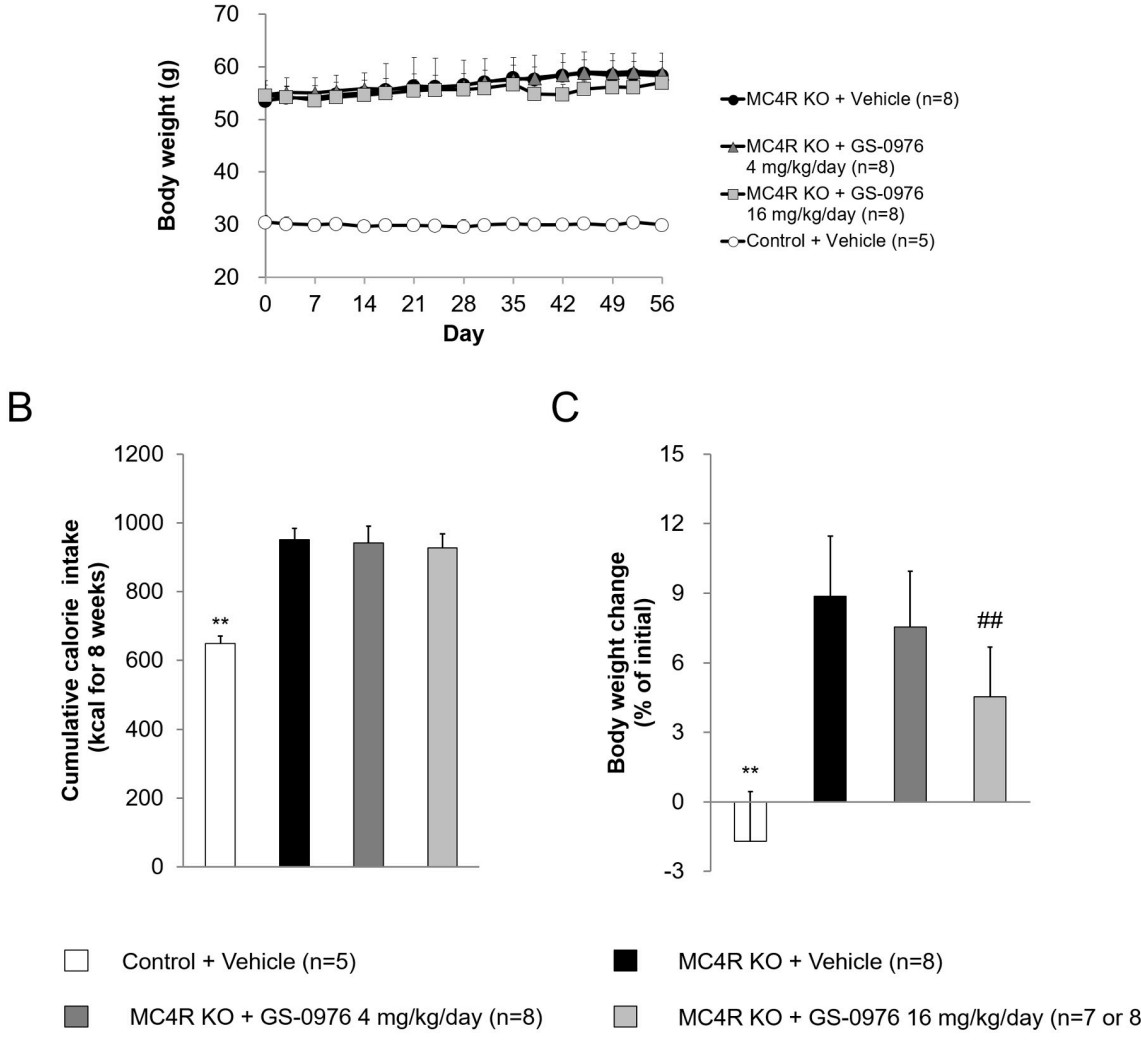

**Fig 4. Effects of GS-0976 on body weight and cumulative calorie intake.** (A) Change of body weight for 8 weeks. (B) Cumulative calorie intake for 8 weeks. (C) Rate of body weight change from the values before the start of drug administration. Data are represented as the mean + SD, **p<0.01 vs. MC4R KO mice treated with vehicle by Student's *t*-test. ##p<0.005 vs. MC4R KO mice treated with vehicle by one-tailed Williams' test.

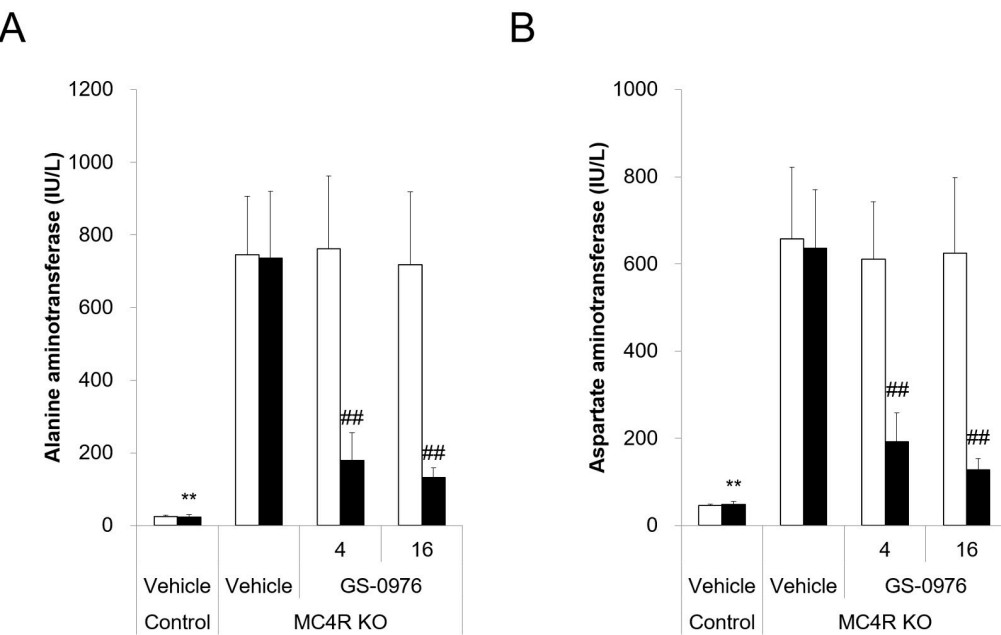

**Fig 5. Effects of GS-0976 on plasma alanine aminotransferase and aspartate aminotransferase levels.** (A) Plasma alanine aminotransferase levels. (B) Plasma aspartate aminotransferase levels. Plasma was obtained before and after the 8-week treatment. The hollow column shows values pre-treatment and the shaded column shows values post-treatment for the lean control group (Control, n = 5) and MC4R KO mice treated with vehicle or GS-0976 (4 or 16 mg/kg/day) (n = 8). Data are represented as the mean + SD, **p<0.01 vs. MC4R KO mice treated with vehicle by Aspin-Welch test. ##p<0.005 vs. MC4R KO mice treated with vehicle by one-tailed Shirley-Williams test.

weights and triglyceride content in WD-fed MC4R KO mice before and after treatment was similar (liver weight before: 6.52 ± 0.82, after: 6.56 ± 0.84 g; triglyceride content before: 116 ± 11, after: 105 ± 15 mg/g tissue), suggesting that hepatic steatosis had already been established before drug treatment by pre-feeding with WD for 13 weeks. Liver weights in the GS-

**Table 1. Effects of GS-0976 on plasma glucose, triglyceride, total cholesterol, and insulin concentrations.**

| Mice | | Control | MC4R KO | | |
|---|---|---|---|---|---|
| Treatment | | Vehicle | Vehicle | GS-0976 | |
| Dose (mg/kg/day) | | | | 4 | 16 |
| | | (n = 5) | (n = 8) | (n = 8) | (n = 8) |
| Total cholesterol (mg/dL) | Pre | 91 ± 4 | 363 ± 35 | 373 ± 42 | 371 ± 48 |
| | Post | 85 ± 6** | 389 ± 39 | 264 ± 37## | 249 ± 28## |
| Triglyceride (mg/dL) | Pre | 111 ± 10 | 96 ± 23 | 94 ± 19 | 106 ± 29 |
| | Post | 86 ± 26 | 90 ± 35 | 170 ± 47## | 152 ± 54## |
| Glucose (mg/dL) | Pre | 146 ± 7 | 140 ± 6 | 133 ± 10 | 133 ± 13 |
| | Post | 149 ± 9 | 143 ± 16 | 163 ± 15# | 173 ± 18## |
| Insulin (ng/mL) | Pre | 0.9 ± 0.4 | 12.4 ± 5.8 | 13.7 ± 7.7 | 16.2 ± 6.3 |
| | Post | 0.7 ± 0.1** | 9.4 ± 3.9 | 59.9 ± 39.8## | 79.3 ± 22.7## |

Plasma parameters were measured in lean control mice (Control, n = 5) and MC4R KO mice (n = 8) before (Pre) and after 8-week treatment (Post). Data are represented as the mean ± SD,

**p<0.01 vs. MC4R KO mice treated with vehicle by Aspin-Welch test.

#p<0.025,

##p<0.005 vs. MC4R KO mice treated with vehicle by one-tailed Williams' test or one-tailed Shirley-Williams test.

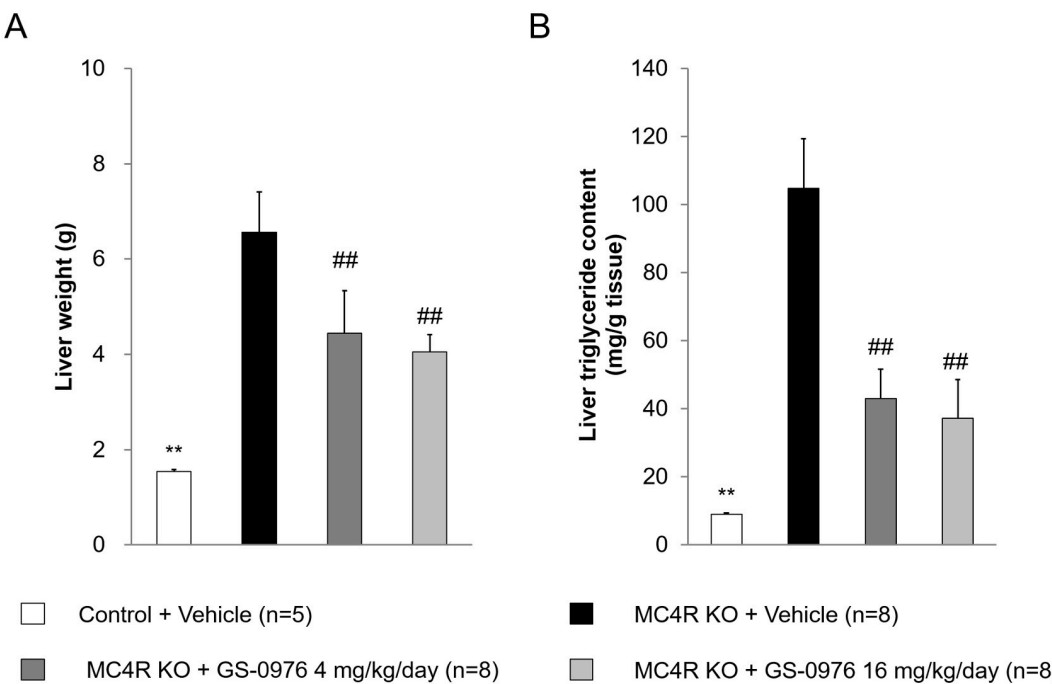

**Fig 6. Effects of GS-0976 on liver weight and liver triglyceride content.** (A) Liver weight. (B) Liver triglyceride content. The liver was harvested one hour after the last administration of the 9-week treatment. Data are represented as the mean + SD, $^{**}$p<0.01 vs. MC4R KO mice treated with vehicle by Aspin-Welch test, $^{##}$p<0.005 vs. MC4R KO mice treated with vehicle by one-tailed Williams' test.

0976-treated groups (4 and 16 mg/kg/day) were significantly lower compared to those of vehicle-treated group by 32 and 38%, respectively. Similarly, hepatic triglyceride content was also significantly lower by 59 and 65% in GS-0976-treated groups. These data demonstrate that GS-0976 has therapeutic effects on hepatic steatosis in WD-fed MC4R KO mice.

## NAFLD activity score

Histological analysis demonstrated an increase of the hepatic steatosis and weak lobular inflammation in vehicle-treated MC4R KO mice fed with WD after 9-week treatment compared with lean control mice (Fig 7, Table 2). On the other hand, ballooning degeneration was not clearly observed in vehicle-treated MC4R KO mice, suggesting that WD-fed MC4R KO mice showed severe steatosis with weak inflammation in the liver. Treatment with GS-0976 at 4 and 16 mg/kg/day lowered the steatosis score dose-dependently but did not show clear effect on inflammation score compared with vehicle treatment. As a result, GS-0976 lowered NAFLD activity score dose-dependently compared with vehicle treatment, reflecting the reduction of the steatosis score by GS-0976.

## Hepatic gene expression related to fibrosis and inflammation

NASH patients exhibit fibrosis and inflammation in the liver, which are hypothesized to be induced by hepatic steatosis. Therefore, we evaluated the mRNA expression of *Col1a1*, *Col1a2*, and *TGF-β1* as markers of fibrosis and *F4/80* as a marker of macrophage in the liver. The mRNA levels of *Col1a1*, *Col1a2*, *TGF-β1*, and *F4/80* were significantly higher in the liver of MC4R KO mice compared with control mice (Fig 8). The mRNA levels of *Col1a1* were

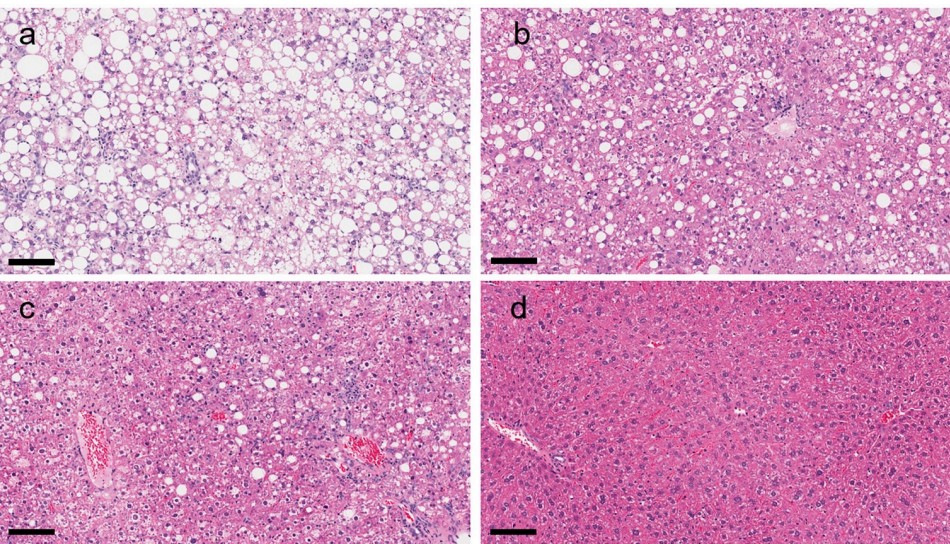

**Fig 7. Effects of GS-0976 on NAFLD activity score in histological analysis.** Histological sections of the liver from vehicle group (a), GS-0976 4 mg/kg/day group (b), GS-0976 16 mg/kg/day group (c) and lean control group (d). Hematoxylin and eosin, bar = 100 μm.

significantly reduced by treatment with GS-0976 at 4 and 16 mg/kg/day, and the levels of the *Col1a2* gene were also significantly reduced by GS-0976 at 16 mg/kg/day. In contrast, GS-0976 did not change the expression levels of *F4/80* and *TGFβ1* genes.

## Hepatic hydroxyproline content, fibrosis area and plasma TIMP-1

We examined direct effect of GS-0976 on fibrosis by measurement of hepatic hydroxyproline content and Sirius Red-positive area for hepatic collagen deposition. Hepatic hydroxyproline content in MC4R KO mice was 4.1-fold higher than that in control mice at the end of the study (Fig 9C). Hepatic hydroxyproline content increased 1.9-fold with the 9-week WD feeding in MC4R KO mice, suggesting that fibrosis had developed before drug treatment and was exacerbated during the 9-week WD feeding (before: 1.4 ± 0.4 mg/g tissue, after: 2.6 ± 0.4 mg/g tissue). GS-0976 at 4 and 16 mg/kg/day significantly lowered hydroxyproline content

**Table 2. Effects of GS-0976 on steatosis, lobular inflammation, ballooning degeneration, and NAFLD activity score.**

| Mice | Control | MC4R KO | | |
|---|---|---|---|---|
| Treatment | Vehicle | Vehicle | GS-0976 | |
| Dose (mg/kg/day) | | | 4 | 16 |
| | (n = 5) | (n = 8) | (n = 8) | (n = 8) |
| Steatosis | 0.0 ± 0.0 | 3.0 ± 0.0 | 1.9 ± 0.4 | 1.3 ± 0.5 |
| Lobular inflammation | 0.0 ± 0.0 | 1.0 ± 0.0 | 0.9 ± 0.4 | 1.0 ± 0.0 |
| Ballooning degeneration | 0.0 ± 0.0 | 0.0 ± 0.0 | 0.0 ± 0.0 | 0.0 ± 0.0 |
| NAFLD activity score | 0.0 ± 0.0 | 4.0 ± 0.0 | 2.8 ± 0.7 | 2.3 ± 0.5 |

Individual scores for steatosis (0–3), inflammation (0–3) and ballooning (0–2) were provided and were added up to determine the NAFLD activity score as a semi-quantitative measure of disease severity. Steatosis, inflammation and hepatocyte ballooning were scored according to the previous report [33].

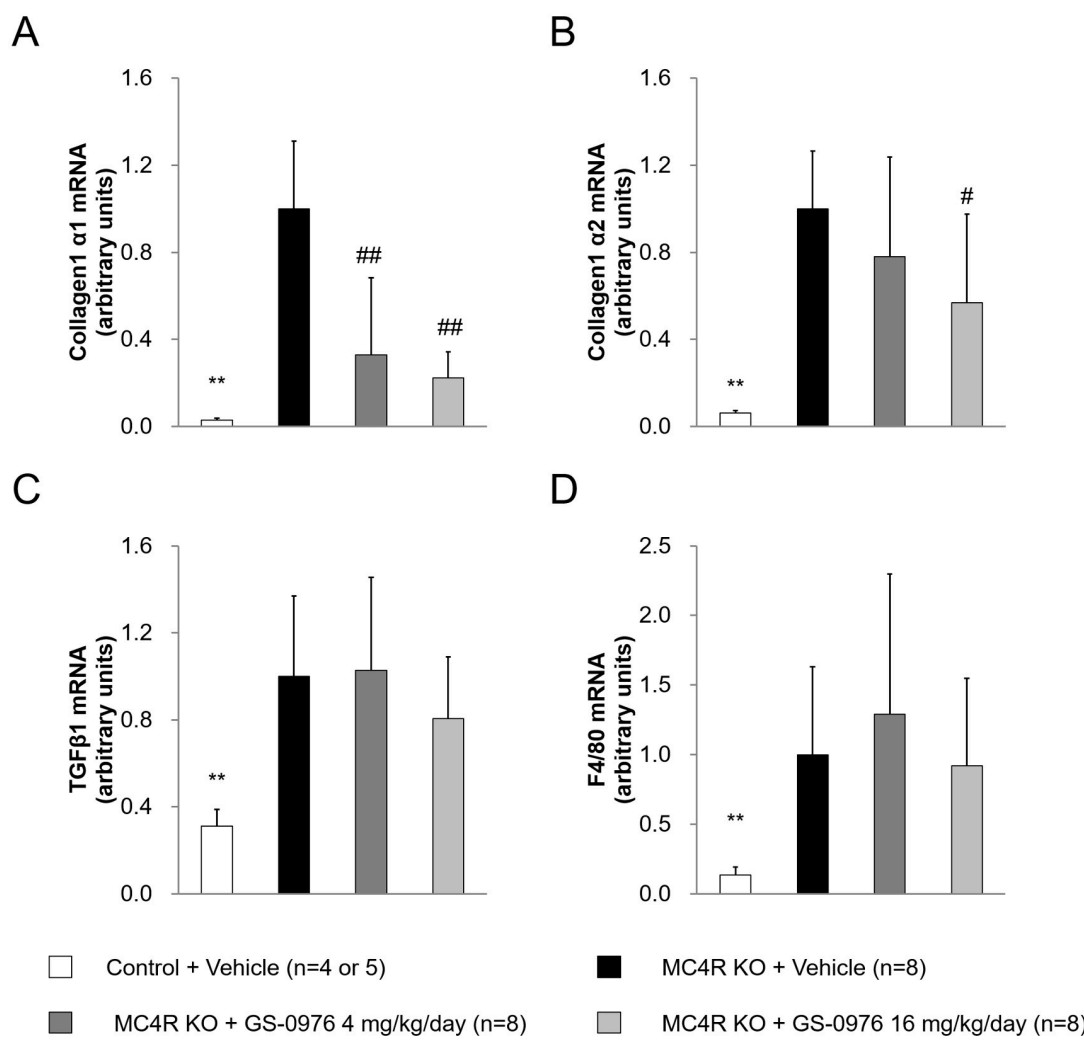

**Fig 8. Effects of GS-0976 on hepatic mRNA expression.** Gene expressions of (A) collagen1 α1 (*Col1a1*), (B) collagen1 α2 (*Col1a2*), (C) Transforming growth factor β1 (*TGFβ1*) and (D) *F4/80*. The liver was harvested one hour after the last administration of the 9-week treatment. Gene expressions were measured in the liver homogenates. Data are represented as the mean + SD, **p<0.01 vs. MC4R KO mice treated with vehicle by Aspin-Welch test, #p<0.025, ##p<0.005 vs. MC4R KO mice treated with vehicle by one-tailed Williams' test or Shirley-Williams test.

compared to vehicle treatment by 28 and 35%, respectively (4 mg/kg/day: 1.9 ± 0.6, 16 mg/kg/day: 1.7 ± 0.5 mg/g tissue). Since GS-0976 did not completely decrease hydroxyproline content to pre-drug treatment levels, GS-0976 suppressed the progression of fibrosis in WD-fed MC4R KO mice. Furthermore, GS-0976 at 4 and 16 mg/kg/day also significantly decreased the Sirius Red-positive area by 40 and 39%, respectively (Fig 9A and 9B). Fibrosis in the liver is accompanied by extracellular matrix remodeling, and TIMP-1, matrix metalloproteinases inhibitor, play an important role. We measured plasma TIMP-1 concentrations after 8-week treatment. Plasma TIMP-1 concentrations in WD-fed MC4R KO mice were 3.2 times higher compared with those of lean control mice (Fig 9D). Treatment with GS-0976 at 4 and 16 mg/kg/day significantly lowered plasma TIMP-1 concentrations by 49 and 64% compared with vehicle treatment, respectively.

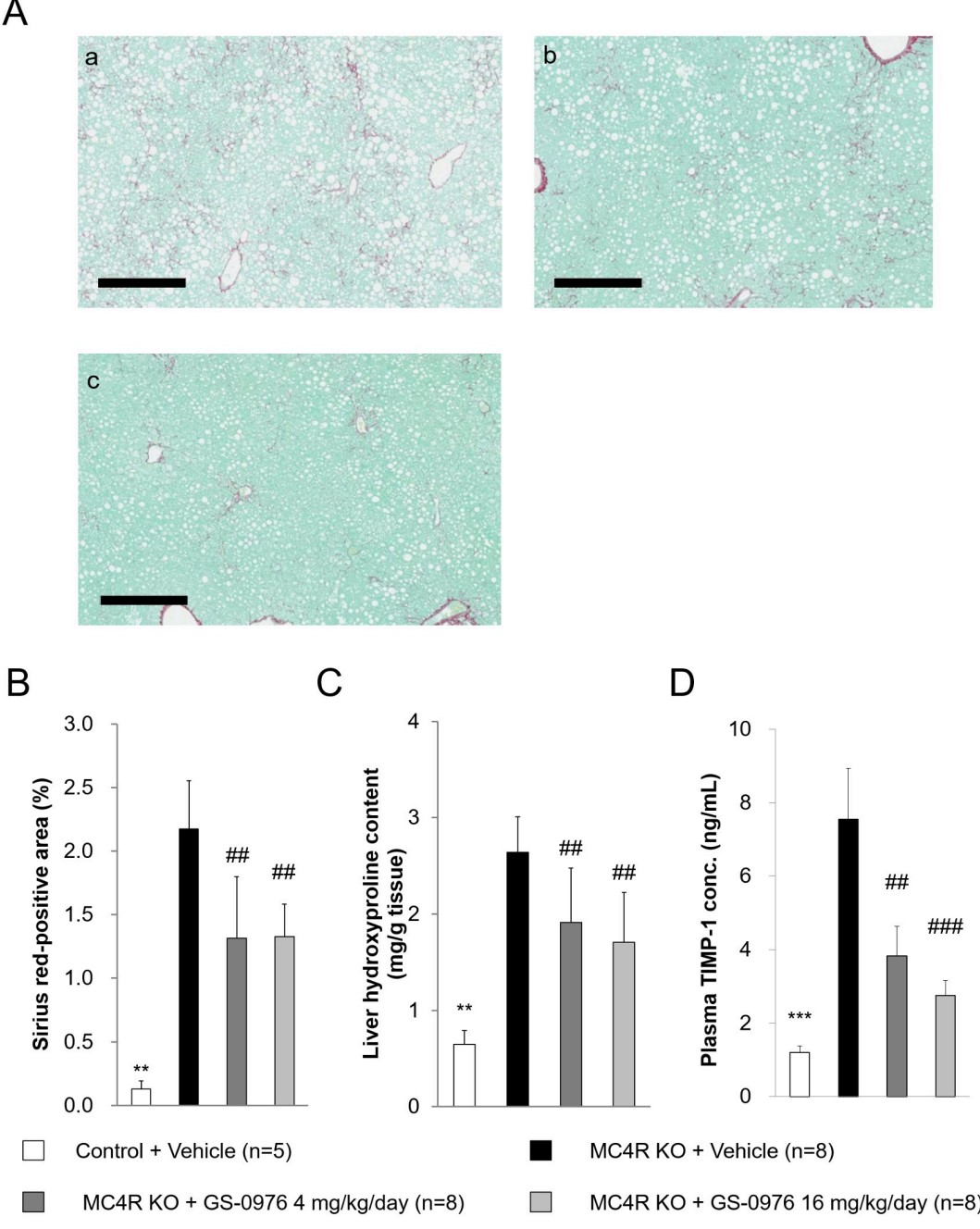

**Fig 9. Effect of GS-0976 on fibrosis.** (A) Representative images of liver sections stained with Sirius Red from vehicle group (a), GS-0976 4 mg/kg/day group (b), and GS-0976 16 mg/kg/day group (c), bar = 400μm. (B) Fibrosis areas in Sirius Red-stained sections were quantified using Scanscope XT. (C) Hydroxyproline content was measured in the liver homogenates as an index of collagen content. (D) Plasma TIMP-1 concentrations were measured after 8-week treatment. Data are represented as the mean + SD, $^{**}$p<0.01 vs. MC4R KO mice treated with vehicle by Aspin-Welch test. $^{##}$p<0.005 vs. MC4R KO mice treated with vehicle by one-tailed Williams' test or Shirley-Williams test.

## Hepatic malonyl-CoA content

Hepatic malonyl-CoA content 1 hour after the last administration at the end of study was measured to confirm the relationship between ACC inhibition and efficacy of GS-0976 in the

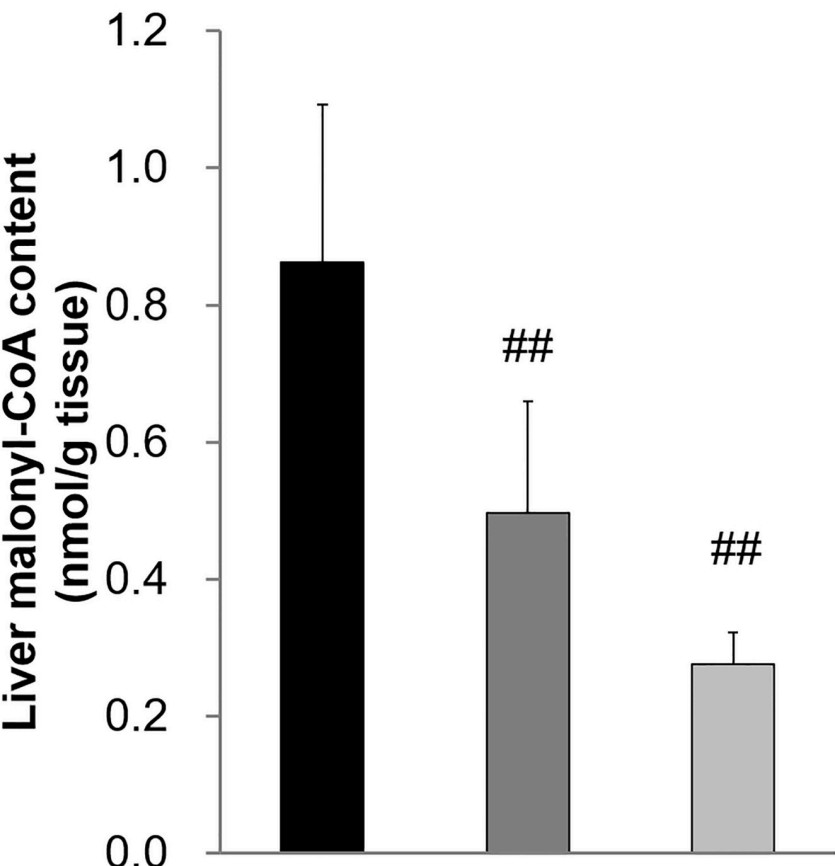

**Fig 10. Effects of GS-0976 on liver malonyl-CoA content in repeated dosing study.** The liver was harvested one hour after the last administration of the 9-week treatment in MC4R KO mice. Data are represented as the mean + SD, ##p<0.005 vs. MC4R KO mice treated with vehicle by one-tailed Shirley-Williams test.

NASH model. GS-0976 at 4 and 16 mg/kg/day lowered hepatic malonyl-CoA content by 43 and 68%, respectively, indicating that GS-0976 inhibited ACC in the liver in WD-fed MC4R KO mice (Fig 10).

## Discussion

Our study demonstrated that WD-fed MC4R KO mice had many characteristics in common with NASH patients, such as hepatic steatosis, increased Sirius Red-positive area, hepatic hydroxyproline content, fibrosis-related gene expression in the liver and plasma liver enzyme levels as reported previously [29, 30]. In this model, GS-0976, a liver-targeted inhibitor of both ACC1/2, significantly improved severe steatosis, suppressed the progression of fibrosis, and reduced plasma liver enzyme levels.

It is known that ACC1/2 dual inhibition reduces hepatic triglyceride content both in normal rodents [20, 24] and in rats with diet-induced obesity [25]. Furthermore, MK-4074, another potent liver targeted inhibitor of ACC1/2, reduced DNL with both single doses and 7-day treatments in healthy subjects; it improved hepatic steatosis with 4 weeks of treatment [20]. Our data also showed that GS-0976 at 4 and 16 mg/kg/day robustly lowered hepatic triglyceride content and improved steatosis histologically in WD-fed MC4R KO mice (Figs 6B and 7). Since GS-0976 at these doses suppressed liver malonyl-CoA content (Fig 10), indicating that it improved hepatic steatosis through the suppression of DNL via ACC1/2 inhibition. Since hepatic DNL in patients with NAFLD is higher compared with that of healthy volunteers [18, 19, 34], the inhibition of DNL by GS-0976 might substantially contribute to a reduction in hepatic lipids. Another potential reason for the massive reduction of hepatic triglyceride content due to GS-0976 is the enhancement of fatty acid oxidation in the liver by ACC2 inhibition. Suppression of ACC2 with antisense oligonucleotides increased fatty acid oxidation in hepatocytes, and ACC1/2 dual inhibition by the same method further increased fatty acid oxidation compared with inhibition of ACC2 alone [24]. Dual inhibition of ACC1/2 could improve hepatic steatosis in patients with NASH through both suppression of DNL and increased fatty acid oxidation.

Fibrosis progression in patients with NASH increases the risks of liver-related morbidity and mortality because cirrhosis develops from the progression of fibrosis and can increase the risk of hepatocellular carcinoma. Improving fibrosis or suppressing its progression would be a desired outcome for intervention [35]. In our study, histological fibrosis area, hydroxyproline content, and collagen mRNA expression in the liver were increased in WD-fed MC4R KO mice, and GS-0976 suppressed these fibrosis parameters. Furthermore, GS-0976 decreased plasma TIMP-1 in WD-fed MC4R KO mice (Fig 9D). Recent clinical study has shown that GS-0976 reduced serum TIMP-1 in patients with NAFLD [27, 36], suggesting translation of the anti-fibrosis effect from liver-targeted ACC1/2 dual inhibition in NASH patients. Although GS-0976 suppressed the hepatic mRNA expression of *Col1a1* and *Col1a2*, it showed no effect on the mRNA expression of *TGFβ1* which is one of the most important factors in stimulating type I collagen gene transcription [37, 38]. It was reported that eicosapentaenoic acid showed no effect on mRNA expression of *TGFβ1* but suppressed active TGFβ1 protein content in the liver [30]. TGFβ is constitutively synthesized and secreted in a biologically latent form (latent TGFβ), and latent TGFβ is activated through proteolytic cleavage of latency-associated peptide region by serine proteases such as matrix metalloproteinases, plasminogen activators, and αvβ6 integrin cleavage [38, 39]. Therefore, GS-0976 does not affect TGFβ mRNA, but might affect the levels of active TGFβ. In our study, there were the lack of dose-dependent response to GS-0976 in Sirius red-positive area and small differences between two doses in some parameters, although we determined doses of GS-0976 based on the influence on PD marker in single dosing study (Fig 2). GS-0976 also showed a dose-dependent inhibitory effect on PD markers in the repeated study (Fig 10). This might be because the histological evaluation is usually evaluated using a single section. Furthermore, it was also possible that low dose was enough to show efficacy on steatosis and fibrosis in this model.

In our model, fibrosis is thought to be indirectly induced through the accumulation of increased fat in the liver. Saturated free fatty acids such as palmitate and stearate, final products of DNL, and their metabolites contribute to lipotoxicity, hepatocyte injury and lipoapoptosis, leading to fibrosis [40]. Oxidative stress is an important factor for inducing fibrosis in NASH. In hepatocytes, oxidative stress is induced by hepatic microsomal lipid peroxidation due to excessive fatty acid delivery and electron leakage from the mitochondrial electron transport system [41, 42]. Inhibition of excess lipogenesis by GS-0976 may contribute to reduction of oxidative stress, resulting in the prevention of fibrosis. Evaluation of oxidative stress in MC4R

KO mice treated with GS-0976 is an area of future research. Inflammation is another factor promoting fibrosis in NASH. Gene expressions of F4/80, monocyte chemoattractant protein 1 (MCP-1) and tumor necrosis factor α (TNFα), not but IL-6 were upregulated in WD-fed MC4R KO mice compared with control mice and upregulations of MCP-1 and TNF α mRNA were inhibited by GS-0976 (Fig 8, S1 Fig). However, histological analysis revealed that the level of inflammatory cell infiltration in GS-0976-treated MC4R KO mice was not different from that in vehicle-treated MC4R KO mice (Table 2). This is probably because that the effect of GS-0976 could not be detected due to weak inflammation observed in the model rather than poor efficacy of GS-0976 against inflammation, because lobular inflammation score was low (1.0 ± 0.0) in vehicle-treated MC4R KO mice. The increase in mRNA expression of cytokines might not reach the level that causes infiltration of inflammatory cells.

ACC1 homozygous knockout mice do not survive as embryos [43], while ACC2 homozygous knockout mice are healthy and fertile [44], suggesting that ACC inhibition, especially with ACC1, might induce unexpected side effects if the inhibition takes place in the whole body. However, GS-0976 did not cause any abnormal findings except for some increases in blood biochemistry parameters in our study. GS-0976 is liver specific because it was designed to be a substrate of hepatic organic anion-transporting polypeptide, resulting in liver-directed biodistribution and ensuring inhibition of ACC in the liver [26]. GS-0976 caused a significant reduction of malonyl-CoA in the liver, at a dose 10-fold lower than that effective in skeletal muscle in rats [25]. This high specificity to the liver could be contributing to low adverse effects. These insights lead us to expect that liver-targeted GS-0976 could be a safer therapeutic agent for NASH than conventional ACC1 and ACC2 inhibitors without organ specificity.

On the other hand, GS-0976 significantly increased plasma triglyceride concentrations in our study (Table 1). Consistent with these results, liver specific double knockout of ACC1/2 also demonstrated increased plasma triglyceride concentrations in mice fed with normal chow or WD [20]. Recently, it was reported that both GS-0976 and MK-4074 increased plasma triglyceride levels in patients with nonalcoholic fatty liver disease, consistent with our pre-clinical data [20, 31]. It was demonstrated that suppression of DNL decreased polyunsaturated fatty acid (PUFA) content as well as saturated fatty acid content in the liver, and reduction of PUFA content increased the expressions of both SREBP-1c and its downstream target glycerol-3-phosphate acyltransferase 1, leading to increased secretion of triglyceride as very low density lipoprotein from the liver [20]. Therefore, plasma triglyceride elevation might be an on-target effect by ACC inhibition. However, others have reported that GS-0976 reduced plasma triglyceride levels in obese rats fed with a high-sucrose diet and that GS-0976 did not affect serum triglyceride in NASH patients [25, 36]. In addition to plasma triglyceride, plasma glucose and insulin levels were also increased by GS-0976 in our study (Table 1). The exact mechanism is unclear, but an increase in gluconeogenesis may contribute to these phenomena. GS-0976 improve fatty liver but may increase acetyl-CoA content in the liver by inhibition of ACC1/2. In hepatocytes, it has been reported that gluconeogenesis from lactic acid is increased by acetate, which is a substrate for acetyl-CoA, therefore gluconeogenesis could be promoted by increasing acetyl-CoA [45, 46]. Furthermore, increased hepatic acetyl-CoA by GS-0976 may have promoted gluconeogenesis and caused increase in plasma glucose and compensatory plasma insulin. On the other hand, GS-0976 lowered plasma insulin concentrations and exhibited no effects on plasma glucose concentrations in rats with a high-fat diet-induced obesity [25]. Furthermore, GS-0976 did not increase plasma glucose, insulin levels or induce insulin resistance in NASH patients [25, 36]. These differences may be due to the severity of insulin resistance. Further investigation is needed to clarify the mechanisms of elevation of plasma triglyceride, glucose and insulin levels by GS-0976 in WD-fed MC4R KO mice and further understanding will be elucidated as additional clinical trial results are disclosed.

Although the MC4R KO mice used in this study were independently created in our laboratory, the phenotype is similar with that of the WD-fed MC4R KO mice and reflects multiple aspects of pathophysiology of NASH patients such as liver injury, steatosis, and fibrosis [29]. NASH develops from metabolic disorders such as insulin resistance and obesity [47, 48]. In WD-fed MC4R KO mice, obesity and systemic insulin resistance are likely induced by the combination of hyperphagia induced by MC4R deficiency and dietary lipids and fructose [49, 50]. Furthermore, it was reported that the activation of lipogenesis, which is the physiological action of insulin in the liver, remains in animal models fed high fat diets [51] and in NAFLD patients [52] despite systemic insulin resistance. Hyperinsulinemia in WD-fed MC4R KO mice might further accelerate the development of steatosis. This animal model is considered a suitable NASH model based on obesity and insulin resistance. In contrast to WD-fed MC4R KO mice, diet-induced and chemically induced models using normal mice do not fully reflect human NASH pathology. Even though a high fat-diet supplemented with fructose or sucrose elicit obesity, insulin resistance, steatosis and steatohepatitis, fibrosis was not observed or mild in nature [50]. Combining WD with MC4R KO mice, which induces substantial obesity and insulin resistance compared to the high fat diet-fed normal mice, is an attractive NASH model with significant fibrosis. Nutrient-deficient diets which are low or devoid of methionine and/or choline are applied to induce severe liver fibrosis. Chemically induced liver damage models are also used for studying mechanisms of hepatic fibrosis progression. However, both nutrient-deficient models and chemically induced models do not fully reflect human NASH pathology, because these models show weight loss [40, 53]. From these reasons, WD-fed MC4R KO mice could be a better pre-clinical model to study the pharmacology of potential NASH therapies compared to nutrient- and chemical-induced models.

In conclusion, we demonstrated that ACC1/2 liver-targeted dual inhibition not only improved hepatic steatosis but also suppressed fibrosis progression in WD-fed MC4R KO mice with severe hepatic steatosis and fibrosis. However, our results also suggested potential risks of GS-0976 as we observed abnormalities in glucose and lipid metabolism. Liver-targeted ACC1/2 inhibitors would be promising drugs for NASH patients although attention must be paid to systemic effects on glucose and lipid metabolism in clinical use.

## Supporting information

**S1 Fig.**
(PPTX)

**S1 Table.**
(PPTX)

## Acknowledgments

The authors thank our lab members for their cooperation throughout this study. The authors also thank Drs. Miho Imawaka, Kazuya Kawasaki, Hitoshi Kandori and Ryotaro Hori for clinical observation and histopathological examination.

## Author Contributions

**Conceptualization:** Hiroaki Yashiro.

**Data curation:** Mitsuharu Matsumoto, Manami Kaneko.

**Investigation:** Mitsuharu Matsumoto, Hitomi Ogino, Kazunobu Aoyama, Tadahiro Nambu, Sayuri Nakamura, Mayumi Nishida, Manami Kaneko.

**Project administration:** Hiroaki Yashiro, Xiaolun Wang, Derek M. Erion.

**Supervision:** Hiroaki Yashiro, Derek M. Erion, Manami Kaneko.

**Writing – original draft:** Mitsuharu Matsumoto.

**Writing – review & editing:** Hiroaki Yashiro, Derek M. Erion, Manami Kaneko.

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
