## [Decision Letter · Decision Letter 0]

21 Aug 2019

PONE-D-19-19866

Acetyl-CoA carboxylase 1 and 2 inhibition ameliorates steatosis and hepatic fibrosis in a MC4R knockout murine model of nonalcoholic steatohepatitis

PLOS ONE

Dear Matsumoto,

Thank you for submitting your manuscript to PLOS ONE. After careful consideration, we feel that it has merit but does not fully meet PLOS ONE’s publication criteria as it currently stands. Therefore, we invite you to submit a revised version of the manuscript that addresses the points raised during the review process.

We would appreciate receiving your revised manuscript by Oct 05 2019 11:59PM. To enhance the reproducibility of your results, we recommend that if applicable you deposit your laboratory protocols in protocols.io, where a protocol can be assigned its own identifier (DOI) such that it can be cited independently in the future. For instructions see: http://journals.plos.org/plosone/s/submission-guidelines#loc-laboratory-protocols

We look forward to receiving your revised manuscript.

Kind regards,

Nobuyuki Takahashi, Ph.D.

Academic Editor

PLOS ONE

Journal requirements;

1. We note that you have included the phrase “data not shown” in your manuscript. Unfortunately, this does not meet our data sharing requirements. PLOS does not permit references to inaccessible data. We require that authors provide all relevant data within the paper, Supporting Information files, or in an acceptable, public repository. Please add a citation to support this phrase or upload the data that corresponds with these findings to a stable repository (such as Figshare or Dryad) and provide and URLs, DOIs, or accession numbers that may be used to access these data. Or, if the data are not a core part of the research being presented in your study, we ask that you remove the phrase that refers to these data.

Reviewers' comments:

Reviewer's Responses to Questions

**Comments to the Author**

1. Is the manuscript technically sound, and do the data support the conclusions?

Reviewer #1: Yes

Reviewer #2: No

2. Has the statistical analysis been performed appropriately and rigorously? 

Reviewer #1: Yes

Reviewer #2: Yes

3. Have the authors made all data underlying the findings in their manuscript fully available?

Reviewer #1: No

Reviewer #2: Yes

4. Is the manuscript presented in an intelligible fashion and written in standard English?

Reviewer #1: Yes

Reviewer #2: Yes

5. Review Comments to the Author

Reviewer #1: In this study, Matsumoto et al. examined the effect of the ACC1/2 inhibitor ND-630 on the progression of NASH using a murine model of NASH with similar phenotypes to human NASH patients. The authors showed that ND-630 treatment effectively ameliorated hepatic steatosis along with liver fibrosis in the NASH model. Considering an unmet medical need for the treatment of NASH, this study provides significant information on the therapeutic strategy of NASH targeting de novo lipogenesis in the liver. However, additional experiments and discussion are required to understand the overall effects of ND-630 on the NASH model. Specific comments are described below.

1. First of all, the authors should validate their MC4R KO mice fed WD as a NASH model, because these mice were independently created in this study. Histological evaluation should be carefully performed using the NAFLD activity score. Did they histologically analyze the mice when ND-630 treatment started at 24-weeks of age?

2. The effect of ND-630 on hepatic inflammation should be examined in detail. Did the treatment affect the number of macrophages and expression of proinflammatory cytokines in the liver?

3. It is also important to know the effect of ND-630 on the fibrogenic process. The data on Timp-1 (discussed in page 26) should be provided. Since ND-630 treatment markedly suppressed Col1a1 mRNA expression, without affecting mRNA expression of Col1a2 and Tgfb1, how do the authors discuss the underlying mechanisms?

4. It is interesting that ND-630 treatment remarkably increased serum insulin concentrations, whereas it suppressed hepatic steatosis. Can the authors provide the plausible reason? Did the authors examine the effect of ND-630 on serum insulin concentrations in wild-type mice fed WD?

5. Did ND-630 treatment influence on lipid accumulation and inflammation in other organs, such as adipose tissue and skeletal muscle?

6. It would be intriguing if quality of lipid was analyzed in the liver of ND-630-treated mice.

Reviewer #2: In this work, Matsumoto et al. report that treatment with two oral doses of the small molecule inhibitor of acetyl-CoA carboxylases 1 and 2 ND-613, given for 9 weeks, attenuates some biochemical and histological markers of liver damage in Western diet-fed melanocortin 4 receptor-deficient mice, a postulated model for NAFLD. They employed C57BL/6J mice fed chow as controls for their experiments. Based on previous findings in Zucker diabetic rats treated with the same inhibitor, in which it was observed that the blockade of ACC1/2 decreased hepatic steatosis and a number or markers of hepatic inflammation and fibrosis, Matsumoto et al. hypothesize that similar effects of ACC1/2 inhibition would be observed in a mouse NAFLD model. Although the present manuscript offers evidence, albeit limited, that add to the postulate that ACC1/2 play a role in improving the biochemical and biological phenotype of NAFLD, this work raises the following critical issues:

- A major problem of the present work is the lack of appropriate control groups for the dietary and drug treatments in addition to the lean mice fed chow. There is no control for the mice subjected to the Western diet, and also a proper control group of wild type mice treated with the ACC1/2 inhibitor. Without data from those two additional groups the reported observations could be attributed to differences in diets, or off-target and toxic effects of the ACC 1/2 inhibitor.

- Although there is no optimal experimental model for NAFLD, the authors don’t explain the rationale for employing the melanocortin 4 receptor-deficient mice as a more suitable model than the more commonly used diet-based or chemical NAFLD models. Alterations in the mechanisms of appetite and control of food intake promoted by a dysfunctional melanocortin 4 receptor pathway could be potentially responsible for some of the hormonal and metabolic changes described in this work.

- Treatment with ND-613 causes a severe hyperinsulinemia in melanocortin 4 receptor-deficient, a relevant finding that was not properly discussed in the manuscript. Also important, the lack of dose-dependent response to the ACC inhibitor in some determinations (liver weight, hepatic triglyceride, hydroxyproline content, etc.) was not addressed in the discussion.

6. PLOS authors have the option to publish the peer review history of their article (what does this mean?). If published, this will include your full peer review and any attached files.

Reviewer #1: No

Reviewer #2: No

---

## [Author Response · Author response to Decision Letter 0]

23 Dec 2019

We attached colored word file of "Response to reviewers". Because it is better to find modifications from original manuscript, I would appreciate if you see that file.

Response to Reviewer #1 comments

Reviewer #1 Comments for the Author

In this study, Matsumoto et al. examined the effect of the ACC1/2 inhibitor ND-630 on the progression of NASH using a murine model of NASH with similar phenotypes to human NASH patients. The authors showed that ND-630 treatment effectively ameliorated hepatic steatosis along with liver fibrosis in the NASH model. Considering an unmet medical need for the treatment of NASH, this study provides significant information on the therapeutic strategy of NASH targeting de novo lipogenesis in the liver. However, additional experiments and discussion are required to understand the overall effects of ND-630 on the NASH model. Specific comments are described below.

Authors’ response

We appreciate your time and effort in reviewing this manuscript and for the useful suggestions that improved the quality of our manuscript. We have responded to the best of our abilities for each of your comments. Line numbers described in this file correspond with those in file of "Revised manuscripts (marked-up copy)".

In this revised version, we changed the description of compound name from ND-630 to GS-0976 because GS-0976 is more widely used than the previous name ND-630. 

Comment 1: First of all, the authors should validate their MC4R KO mice fed WD as a NASH model, because these mice were independently created in this study. Histological evaluation should be carefully performed using the NAFLD activity score. Did they histologically analyze the mice when ND-630 treatment started at 24-weeks of age?

Response: We appreciate this useful comment. We performed histological evaluation for the NAFLD activity score using liver samples at 33-weeks of age. Unfortunately, pre-treated samples at 24-weeks of age were not been fixed for histological evaluation. Histological images and the NAFLD activity score were added in figure 7 and table 2. These results suggest that WD-fed MC4R KO mice show severe steatosis with weak inflammation in the liver at 33-weeks of age. On the other hand, ballooning degeneration was not clearly detected. It may be difficult to detect ballooning in the small animal NASH model, rather than in this model. It is reported that ballooning is difficult to distinguish from microvesicular steatosis in small animal NASH models, and correct evaluation is difficult (Hepatology. 2019; 69(5):2241-2257). Furthermore, hepatocyte ballooning was not detected in methionine and choline deficient diet-fed mice model which is often used as a mouse NASH model (Hepatology. 2019; 69(5): 2241-2257). We added these results of histological analysis in line 370 to line 380 in marked-up copy as follows. 

“NAFLD Activity Score

Histological analysis demonstrated an increase of the hepatic steatosis and weak lobular inflammation in vehicle-treated MC4R KO mice fed with WD after 9-week treatment compared with lean control mice (Fig 7, Table 2). On the other hand, ballooning degeneration was not clearly observed in vehicle-treated MC4R KO mice, suggesting that WD-fed MC4R KO mice showed severe steatosis with weak inflammation in the liver. Treatment with GS-0976 at 4 and 16 mg/kg/day lowered the steatosis score dose-dependently but did not show clear effect on inflammation score compared with vehicle treatment. As a result, GS-0976 lowered NAFLD activity score dose-dependently compared with vehicle treatment, reflecting the reduction of the steatosis score by GS-0976.”

Comment 2: The effect of ND-630 on hepatic inflammation should be examined in detail. Did the treatment affect the number of macrophages and expression of proinflammatory cytokines in the liver?

Response: We conducted two additional experiments to investigate the effects of GS-0976 on hepatic inflammation in more detail according to reviewer’s valuable comment. Gene expression analysis of other inflammation-related markers in addition to F4/80 exhibited that gene expressions of monocyte chemoattractant protein 1 (MCP-1) and tumor necrosis factor α (TNFα), not but IL-6 were upregulated in WD-fed MC4R KO mice compared with control mice and these upregulations were inhibited by GS-0976 as shown in supplemental figure 1. However, histological analysis revealed that the level of inflammatory cell infiltration in GS-0976-treated MC4R KO mice did not change in vehicle-treated MC4R KO mice shown in figure 7 and table 2. This is thought to be because inflammation is not strongly induced in this model rather than the lack of efficacy of GS-0976. Lobular inflammation in the NAS activity score was low (1.0 ± 0.0) in vehicle treated MC4R KO mice. Probably, in this model which were fed with WD for 33 weeks, the increase in mRNA expression of cytokines does not reach the level that causes infiltration of inflammatory cells. Continued long-term high-fat diet loading causes inflammatory cell infiltration and GS-0976 may have an inhibitory effect on the increase in inflammation. 

We added these contents about inflammation and mechanism of anti-fibrotic effects to lines 556 to line 576 in marked-up copy. 

“In our model, fibrosis is thought to be indirectly induced through the accumulation of increased fat in the liver. Saturated free fatty acids such as palmitate and stearate, final products of DNL, and their metabolites contribute to lipotoxicity, hepatocyte injury and lipoapoptosis, leading to fibrosis [40]. Oxidative stress is an important factor for inducing fibrosis in NASH. In hepatocytes, oxidative stress is induced by hepatic microsomal lipid peroxidation due to excessive fatty acid delivery and electron leakage from the mitochondrial electron transport system [41, 42]. Inhibition of excess lipogenesis by GS-0976 may contribute to reduction of oxidative stress, resulting in the prevention of fibrosis. Evaluation of oxidative stress in MC4R KO mice treated with GS-0976 is an area of future research. Inflammation is another factor promoting fibrosis in NASH. Gene expressions of F4/80, monocyte chemoattractant protein 1 (MCP-1) and tumor necrosis factor α (TNFα), not but IL-6 were upregulated in WD-fed MC4R KO mice compared with control mice and upregulations of MCP-1 and TNF α mRNA were inhibited by GS-0976 (Fig 8, Supplemental figure 1). However, histological analysis revealed that the level of inflammatory cell infiltration in GS-0976-treated MC4R KO mice was not different from that in vehicle-treated MC4R KO mice (Table 2). This is probably because that the effect of GS-0976 could not be detected due to weak inflammation observed in the model rather than poor efficacy of GS-0976 against inflammation, because lobular inflammation score was low (1.0 ± 0.0) in vehicle-treated MC4R KO mice. The increase in mRNA expression of cytokines might not reach the level that causes infiltration of inflammatory cells.”

Comment 3: It is also important to know the effect of ND-630 on the fibrogenic process. The data on Timp-1 (discussed in page 26) should be provided. Since ND-630 treatment markedly suppressed Col1a1 mRNA expression, without affecting mRNA expression of Col1a2 and Tgfb1, how do the authors discuss the underlying mechanisms?

Response: We measured plasma TIMP-1 levels after 8-week treatment and added the result in figure 9D, and also added the follows in line 429 to line 435. 

“Fibrosis in the liver is accompanied by extracellular matrix remodeling, and TIMP-1, matrix metalloproteinases inhibitor, plays an important role. We measured plasma TIMP-1 concentrations after 8-weeks of treatment. Plasma TIMP-1 concentrations in WD-fed MC4R KO mice were 3.2 times higher compared with those of lean control mice (Fig 9D). Treatment with GS-0976 at 4 and 16 mg/kg/day significantly lowered plasma TIMP1 concentrations by 49 and 64% compared with vehicle treatment, respectively.”

As pointed out by the reviewer, GS-0976 showed no effect on the mRNA expression of TGFβ1 which is one of the most important factors to stimulate type I collagen gene transcription. We added some sentences to make this reason clearer to understand. The followings show that original manuscript in line 473 to line 477 revised manuscript (“clean copy”, not “marked-up copy”) in line 481 to line 485 as follows: 

“TGFβ precursor is activated through proteolytic cleavage of the latency-associated peptide region by serine proteases such as matrix metalloproteinases, plasminogen activators, and αvβ3 integrin cleavage [46].”  “TGFβ is constitutively synthesized and secreted in a biologically latent form (latent TGFβ), and latent TGFβ is activated through proteolytic cleavage of latency-associated peptide region by serine proteases such as matrix metalloproteinases, plasminogen activators, and αvβ6 integrin cleavage [38, 39]. Therefore, GS-0976 does not affect TGFβ mRNA, but might affect the levels of active TGFβ.” 

 GS-0976 significantly suppressed the hepatic mRNA levels of Col1a1 from low dose but significantly suppressed Col1a2 mRNA at only high dose. On the other hand, Sirius red-positive area in histological analysis and hepatic hydroxyproline content were significantly reduced by treatment with low dose of GS-0976. Type I procollagen is a heterotrimer formed from two pro-alpha1(I) chains produced by COL1A1 gene and one pro-alpha2(I) chain produced by the COL1A2 gene. Therefore, suppression of collagen content might be occurred only by suppression of COL1A1 gene expression.

Comment 4: It is interesting that ND-630 treatment remarkably increased serum insulin concentrations, whereas it suppressed hepatic steatosis. Can the authors provide the plausible reason? Did the authors examine the effect of ND-630 on serum insulin concentrations in wild-type mice fed WD?

Response: We are also interested why GS-0976 increased plasma insulin followed by the elevation of plasma glucose. We added following hypothesis about plausible mechanism of these phenomenon. The followings show that original manuscript in line 502 to line 507 revised manuscript (“clean copy”, not “marked-up copy”) in line 541 to line 555 as follows: 

“Still, ND-630 did not increase plasma glucose, insulin levels or induce insulin resistance in NASH patients, and did not increase plasma glucose levels in obese rats fed a high-sucrose diet [25, 42]. As the cause of the differences between these results is unknown, further investigation is needed to clarify the mechanisms of elevation of plasma TG, glucose and insulin levels by ND-630 in WD-fed MC4R KO mice.”“The exact mechanism is unclear, but an increase in gluconeogenesis may contribute to these phenomena. GS-0976 improve fatty liver but may increase acetyl-CoA content in the liver by inhibition of ACC1/2. In hepatocytes, it has been reported that gluconeogenesis from lactic acid is increased by acetate, which is a substrate for acetyl-CoA, therefore gluconeogenesis could be promoted by increasing acetyl-CoA [45, 46]. Furthermore, increased hepatic acetyl-CoA by GS-0976 may have promoted gluconeogenesis and caused increase in plasma glucose and compensatory plasma insulin. On the other hand, GS-0976 lowered plasma insulin concentrations and exhibited no effects on plasma glucose concentrations in rats with a high-fat diet-induced obesity [25]. Furthermore, GS-0976 did not increase plasma glucose, insulin levels or induce insulin resistance in NASH patients [25, 36]. These differences may be due to the severity of insulin resistance. Further investigation is needed to clarify the mechanisms of elevation of plasma triglyceride, glucose and insulin levels by GS-0976 in WD-fed MC4R KO mice and further understanding will be elucidated as additional clinical trial results are disclosed”

We do not have data about effects of GS-0976 on serum insulin concentrations in wild-type mice fed with WD. It was reported that GS-0976 induced lower plasma insulin and unchanged fasting plasma glucose in diet-induced obese rats which show more moderate peripheral insulin resistance compared to the MC4R KO mice fed with WD (Proc Natl Acad Sci U S A. 2016;113(13): E1796-805). Therefore, it is thought that GS-0976 may not induce elevation of serum insulin concentrations in wild-type mice fed with WD.

Comment 5: Did ND-630 treatment influence on lipid accumulation and inflammation in other organs, such as adipose tissue and skeletal muscle?

Response: We did not examine influence of GS-0976 on lipid accumulation and inflammation in other organs, such as adipose tissue and skeletal muscle. However, for the following reasons, GS-0976 probably has no direct effects on lipid accumulation and inflammation in adipose tissue and skeletal muscle. GS-0976 is taken by organic anion transporting polypeptides (OATPs) and preferentially partitioned into the liver (Hepatology. 2018;68, Suppl. 1734). On the other hand, OATPs are not abundantly expressed in the adipose tissue (Physiol Rev. 2015;95(1): 83-123). Furthermore, it is thought that GS-0976 at the doses selected in this study (2 and 8 mg/kg) do not affect in skeletal muscle because GS-0976 at 10 mg/kg selectively decreased malonyl-CoA content in the liver but not in skeletal muscle of normal mice as shown in figure 2. It was also reported that GS-0976 accumulated 50 times in liver compared to skeletal muscle one hour after GS-0976 at 3 and 10 mg/kg were given in the rats fed with a high-fat diet (Proc Natl Acad Sci U S A. 2016;113(13): E1796-805). 

 We modified the sentence to make it easier to understand. The followings show that original manuscript in line 423 revised manuscript (“clean copy”, not “marked-up copy”) in line 518 to line 521 as follows, “ND-630 is a liver-specific inhibitor” GS-0976 is liver specific because it was designed to be a substrate of hepatic organic anion-transporting polypeptide, resulting in liver-directed biodistribution and ensuring inhibition of ACC in the liver [26].” 

Comment 6: It would be intriguing if quality of lipid was analyzed in the liver of ND-630-treated mice.

Response: We appreciate the reviewer's comments. Although it is interesting for us to measure quality of lipid in the liver of GS-0976-treated mice, it is technically difficult for us at present. Another de novo lipogenesis-related enzyme, fatty acid elongase 6 (Elovl6) which is responsible for converting C16 saturated and monounsaturated fatty acids (FAs) into C18 species contributes to obesity-induced insulin resistance by modifying hepatic C16/C18-related FA composition (Nat Med. 2007;13(10):1193-202.). Therefore, there is a possibility that GS-0976 also changes the lipid composition. We consider that their measurement as a potential next step of our study and outside the scope of the current manuscript. 

 

We attached colored word file of "Response to reviewers". Because it is better to find modifications from original manuscript, I would appreciate if you see that file.

Response to Reviewer #2 comments

Reviewer #2 Comments for the Author

In this work, Matsumoto et al. report that treatment with two oral doses of the small molecule inhibitor of acetyl-CoA carboxylases 1 and 2 ND-613, given for 9 weeks, attenuates some biochemical and histological markers of liver damage in Western diet-fed melanocortin 4 receptor-deficient mice, a postulated model for NAFLD. They employed C57BL/6J mice fed chow as controls for their experiments. Based on previous findings in Zucker diabetic rats treated with the same inhibitor, in which it was observed that the blockade of ACC1/2 decreased hepatic steatosis and a number or markers of hepatic inflammation and fibrosis, Matsumoto et al. hypothesize that similar effects of ACC1/2 inhibition would be observed in a mouse NAFLD model. Although the present manuscript offers evidence, albeit limited, that add to the postulate that ACC1/2 play a role in improving the biochemical and biological phenotype of NAFLD, this work raises the following critical issues:

Authors’ response

We appreciate your time and effort in reviewing this manuscript and for very useful suggestions that improved the quality of our manuscript. We have responded to your comments point by point as follows. Line numbers described in this file correspond with those in file of " Revised manuscripts (marked-up copy)".

In this revised version, we changed the description of compound name from ND-630 to GS-0976 because GS-0976 is more widely used than the previous name ND-630.

Comment 1: A major problem of the present work is the lack of appropriate control groups for the dietary and drug treatments in addition to the lean mice fed chow. There is no control for the mice subjected to the Western diet, and also a proper control group of wild type mice treated with the ACC1/2 inhibitor. Without data from those two additional groups the reported observations could be attributed to differences in diets, or off-target and toxic effects of the ACC 1/2 inhibitor.

Response: We appreciate your valuable comments. The present study demonstrated ACC1/2 inhibitor reduced steatosis and fibrosis in WD-fed MC4R KO mice. But as you pointed out, there are no groups with WD and drug treatments in normal mice. With regard to effect of diet, it was reported that Sirius Red-positive area of C57BL/6J mice fed with WD for 24 weeks were only 0.42% (Am J Pathol. 2008;173(4):993-1001), suggesting that WD alone does not cause fibrosis and loading WD in MC4R KO mice is important for the fibrosis observed in this study. Therefore, in order to reduce animal numbers a normal mouse fed with WD group in the study to evaluate the anti-fibrotic effect of GS-0976 was not included. Regarding the toxic effects of GS-0976, it had no effect on food intake and plasma liver enzymes, a liver injury marker, it was rather reduced them compared to vehicle treatment in our study using WD-fed MC4R KO mice. These results are supported by two toxicity studies using rat (Proc Natl Acad Sci U S A. 2016;113(13):E1796-805). Single oral dosing of GS-0976 at 100, 300, or 1,000 mg/kg showed no statistically significant influences on body weight, and showed no adverse effects on hematology, coagulation, or clinical chemistry parameters. Repeat oral dosing of GS-0976 at 60 mg/kg/day for 28 days showed no clinical signs and no changes in body weight, food consumption, hematology, coagulation, or clinical chemistry parameters. More importantly, adverse effects related to GS-0976 were not observed in clinical studies (Gastroenterology. 2018;155(5):1463-1473, Clin Gastroenterol Hepatol. 2018;16(12): 1983-1991). These insights suggest that efficacy of GS-0976 in our study is not depend on toxic effects.

Comment 2: Although there is no optimal experimental model for NAFLD, the authors don’t explain the rationale for employing the melanocortin 4 receptor-deficient mice as a more suitable model than the more commonly used diet-based or chemical NAFLD models. Alterations in the mechanisms of appetite and control of food intake promoted by a dysfunctional melanocortin 4 receptor pathway could be potentially responsible for some of the hormonal and metabolic changes described in this work.

Response: We appreciate and agree with your valuable comments. We added the following sentence about superiority of MC4R KO mice fed with WD compared to other models such as diet-based or chemical NAFLD models in line 624 to line 648. 

“In contrast to WD-fed MC4R KO mice, diet-induced and chemically induced models using normal mice do not fully reflect human NASH pathology. Even though a high fat-diet supplemented with fructose or sucrose elicit obesity, insulin resistance, steatosis and steatohepatitis, fibrosis was not observed or mild in nature [50]. Combining WD with MC4R KO mice, which induces substantial obesity and insulin resistance compared to the high fat diet-fed normal mice, is considered to be an attractive NASH model with significant fibrosis. Nutrient-deficient diets which are low or devoid of methionine and/or choline are applied to induce severe liver fibrosis. Chemically induced liver damage models are also used for studying mechanisms of hepatic fibrosis progression. However, both nutrient-deficient models and chemically induced models do not fully reflect human NASH pathology, because these models show weight loss [40, 51]. From these reasons, WD-fed MC4R KO mice could be a better pre-clinical model to study the pharmacology of potential NASH therapies compared to nutrient- and chemical-induced models.”

Comment 3: Treatment with ND-613 causes a severe hyperinsulinemia in melanocortin 4 receptor-deficient, a relevant finding that was not properly discussed in the manuscript. Also important, the lack of dose-dependent response to the ACC inhibitor in some determinations (liver weight, hepatic triglyceride, hydroxyproline content, etc.) was not addressed in the discussion.

Response: We added following hypothesis about plausible mechanism of these phenomenon. The followings show that original manuscript in line 502 to line 507 revised manuscript (“clean copy”, not “marked-up copy”) in line 541 to line 555 as follows: Furthermore, in addition to adding Table 2, the format of Table 1 is aligned with Table 2. 

“Still, ND-630 did not increase plasma glucose, insulin levels or induce insulin resistance in NASH patients, and did not increase plasma glucose levels in obese rats fed a high-sucrose diet [25, 42]. As the cause of the differences between these results is unknown, further investigation is needed to clarify the mechanisms of elevation of plasma TG, glucose and insulin levels by ND-630 in WD-fed MC4R KO mice.” “The exact mechanism is unclear, but an increase in gluconeogenesis may contribute to these phenomena. GS-0976 improve fatty liver but may increase acetyl-CoA content in the liver by inhibition of ACC1/2. In hepatocytes, it has been reported that gluconeogenesis from lactic acid is increased by acetate, which is a substrate for acetyl-CoA, therefore gluconeogenesis could be promoted by increasing acetyl-CoA [45, 46]. Furthermore, increased hepatic acetyl-CoA by GS-0976 may have promoted gluconeogenesis and caused increase in plasma glucose and compensatory plasma insulin. On the other hand, GS-0976 lowered plasma insulin concentrations and exhibited no effects on plasma glucose concentrations in rats with a high-fat diet-induced obesity [25]. Furthermore, GS-0976 did not increase plasma glucose, insulin levels or induce insulin resistance in NASH patients [25, 36]. These differences may be due to the severity of insulin resistance. Further investigation is needed to clarify the mechanisms of elevation of plasma triglyceride, glucose and insulin levels by GS-0976 in WD-fed MC4R KO mice and further understanding will be elucidated as additional clinical trial results are disclosed”

 We added following explanation about lack of dose-dependent response to GS-0976 in line 548 to line 555.

”In our study, there were the lack of dose-dependent response to GS-0976 in Sirius-red positive area and small differences between two doses in some parameters, although we determined doses of GS-0976 based on the influence on PD marker in single dosing study (Fig 2). GS-0976 also showed a dose-dependent inhibitory effect on PD markers in the repeated study (Fig 10). This might be because the histological evaluation was performed in one section. Furthermore, it was also possible that low dose was enough to show efficacy on steatosis and fibrosis in this model.”

---

## [Decision Letter · Decision Letter 1]

10 Jan 2020

Acetyl-CoA carboxylase 1 and 2 inhibition ameliorates steatosis and hepatic fibrosis in a MC4R knockout murine model of nonalcoholic steatohepatitis

PONE-D-19-19866R1

Dear Dr. Matsumoto,

We are pleased to inform you that your manuscript has been judged scientifically suitable for publication and will be formally accepted for publication once it complies with all outstanding technical requirements.

With kind regards,

Nobuyuki Takahashi, Ph.D.

Academic Editor

PLOS ONE

Additional Editor Comments:

I read your revised manuscript, because only one of the reviewers who reviewed your previous manuscript evaluated the revised manuscript. As a result, I have decided your revised manuscript is acceptable,

Reviewers' comments:

Reviewer's Responses to Questions

**Comments to the Author**

1. If the authors have adequately addressed your comments raised in a previous round of review and you feel that this manuscript is now acceptable for publication, you may indicate that here to bypass the “Comments to the Author” section, enter your conflict of interest statement in the “Confidential to Editor” section, and submit your "Accept" recommendation.

Reviewer #1: All comments have been addressed

2. Is the manuscript technically sound, and do the data support the conclusions?

Reviewer #1: Yes

3. Has the statistical analysis been performed appropriately and rigorously? 

Reviewer #1: Yes

4. Have the authors made all data underlying the findings in their manuscript fully available?

Reviewer #1: Yes

5. Is the manuscript presented in an intelligible fashion and written in standard English?

Reviewer #1: Yes

6. Review Comments to the Author

Reviewer #1: (No Response)

7. PLOS authors have the option to publish the peer review history of their article (what does this mean?). If published, this will include your full peer review and any attached files.

Reviewer #1: No

---

## [Editor Report · Acceptance letter]

17 Jan 2020

PONE-D-19-19866R1 

Acetyl-CoA carboxylase 1 and 2 inhibition ameliorates steatosis and hepatic fibrosis in a MC4R knockout murine model of nonalcoholic steatohepatitis 

Dear Dr. Matsumoto:

I am pleased to inform you that your manuscript has been deemed suitable for publication in PLOS ONE. Congratulations! Your manuscript is now with our production department. 

With kind regards,

on behalf of

Dr. Nobuyuki Takahashi 

Academic Editor

PLOS ONE